# Preparation of Piezo-Resistive Materials by Combination of PP, SEBS and Graphene

**Helga Seyler** [ID]**, Marián A. Gómez-Fatou** [ID] **and Horacio J. Salavagione *** [ID]

Departamento de Física de Polímeros, Elastómeros y Aplicaciones Energéticas, Instituto de Ciencia y Tecnología de Polímeros (ICTP-CSIC), Juan de la Cierva 3, 28006 Madrid, Spain; helgaseyler@yahoo.de (H.S.); magomez@ictp.csic.es (M.A.G.-F.)

**\*** Correspondence: horacio@ictp.csic.es; Tel.: +34-912-587-549

**Abstract:** The use of polyolefins in structural components requires the simultaneous improvement of stiffness and toughness of the matrix, whilst in the case of sensing components during operation, additional functions are needed such as electrical conductivity. However, providing various desired properties without impairing those intrinsic to the materials can be somewhat challenging. In this study we report the preparation of an isotactic polypropylene (iPP)/styrene–ethylene–butylene–styrene triblock copolymer (SEBS)/graphene system that combines enhanced mechanical properties with electrical conductivity. Blends were prepared by solution mixing (SoM) and solution/solid state mixing (SoM/SSM) formulation routes prior to melt processing. The nanocomposites were characterized by scanning electron microscopy (SEM) and thermogravimetric analysis (TGA) and the electrical and mechanical properties were evaluated. The materials prepared via the SoM/SSM route displayed good electrical conductivity while retaining the mechanical properties of iPP, making them attractive materials for low cost and high throughput structural components with sensing capacity.

**Keywords:** nanocomposites; polyolefin; elastomer; young's modulus; toughness; electrical conductivity

---

## 1. Introduction

Polyolefins (polyethylene and polypropylene) are the most widely used synthetic polymers due to their easy preparation, low cost, good processability, recyclability, and biocompatibility that make them attractive candidates for very different areas like packaging, consumer goods, biomedical, or automotive sectors. During the last decades the incorporation of nanofillers has expanded the versatility of polyolefin-based materials to more advanced and engineering applications [1,2]. Graphene is nowadays one of the most interesting nanomaterials incorporated into polymer matrices due to its exceptional properties [3–7]. The combination of graphene with polyolefins provides materials with additional functionalities like electrical conductivity as well as enhanced mechanical, thermal, or barrier properties, thus broadening their spectrum of applications [2,8–15].

Polypropylene (PP) is increasingly used in the automotive sector. In this particular industry, stiffness and toughness have to be simultaneously improved for structural components that, in the case of PP, is an important challenge [16,17]. One of the main strategies used to improve the impact resistance of PP is based on blending it with thermoplastic elastomers such as styrene–ethylene–butylene–styrene (SEBS) because of its excellent toughness, at the expense of other mechanical properties like tensile strength [18,19]. In order to achieve simultaneous improvements in stiffness and toughness in isotactic polypropylene (iPP), the addition of a third component such as inorganic nanofillers, other crystalline polymers, or additives has been widely explored [20–22].

Thus, the preparation of ternary systems by incorporating SEBS and graphene into PP has emerged as a very promising alternative, not only for structural materials but also for electrically conductive materials which are able to sense changes in the components during operation. In immiscible polymer blends, the distribution and location of the nanofiller is critical which can affect the blends properties, especially the electrical conductivity [23,24]. A selective localization of the nanofiller can tune the performance of the material. In the particular case of simultaneous increments in stiffness and toughness in these ternary systems, the aim is to disperse the rigid filler preferentially in the iPP phase without altering the elastomeric phase.

It has been observed that in ternary PP/SEBS/montmorillonite nanocomposites the rigid filler locates preferentially in the SEBS domains, counteracting the positive effect on the toughness of the polymer [21]. Moreover, the incorporation of graphene into elastomers has led to a significant improvement in Young's modulus and a decrease in the elongation at break with increasing graphene content [25].

The chemical functionalization of the filler has been demonstrated to be a powerful tool to enhance the filler/polymer physical interaction to attain good filler dispersion in the matrix [5,6,26]. Very recently, we demonstrated that graphene can be selectively located into specific SEBS domains by specific functionalization of graphene with short brushes of polyethylene [27,28] or polystyrene [28]. Furthermore, we have also developed a protocol to functionalize graphene with short polypropylene brushes that improves the load sharing with the iPP matrix [10,29].

In this paper, we studied the preparation of iPP/SEBS/graphene nanocomposites by evaluating two different preparation procedures. In addition, two different nanofillers (non-functionalized and polypropylene-modified graphene) were evaluated in order to investigate whether the selective location of graphene had an effect on the final properties of the blends.

## 2. Experimental Section

### 2.1. Materials

iPP (95% isotactic, viscosity average molecular weight of 179,000 g/mol, and polydispersity index of 4.77) was supplied by Repsol (Madrid, Spain). Graphene (G, 1–2 layers; lateral dimensions: $22 \pm 5$ µm, $9 \pm 2$ µm) was purchased from Avanzare Nanotechnology (Navarrete, Spain). The SEBS employed was provided by Dynasol (Madrid, Spain). This elastomer contains 30 wt.% of styrene units and the following molecular weight characteristics as determined by GPC: average molecular weight Mw = 85,000 g/mol; polydispersity index = 1.45. The grafting of polypropylene chains to graphene (GPP) was carried out through acylation chemistry following a previously described procedure [29]. N-methyl-2-pyrrolidinone (NMP) and xylene (mixture of isomers) were purchased from Sigma Aldrich.

### 2.2. Preparation of the Nanocomposites

G and GPP were exfoliated in xylene (7 mg/mL) using a sonicating probe (40%, 1 cycle, 30 min) in an ice bath. Two compounding strategies were designed to prepare highly homogeneous nanocomposites with the optimal matrix/filler interactions and mechanical properties. Thus, the preparation of the nanocomposites was conducted by solution mixing (SoM) and solution/solid state mixing (SoM/SSM) formulation routes followed by melt processing (Figure 1). For simplicity, SoM and SoM/SSM are designed as R1 and R2, respectively (Figure 1).

R1 involved the in-situ solution blending of the iPP, SEBS, and filler in warm xylene. Briefly, iPP and SEBS were dissolved in xylene under reflux conditions and the exfoliated G or GPP dispersion was added to the warm polymer solution under inert atmospheric conditions. The mixture was stirred for an additional 2 h, precipitated in methanol, filtered, and dried overnight in a vacuum desiccator at 60 °C. On the other hand, R2 was a two-step procedure. Briefly, iPP was dissolved in xylene under reflux conditions in an inert atmosphere and the G or GPP dispersion was added to the warm polymer solution under inert atmosphere. The mixture was stirred for an additional 2 h,

precipitated in methanol, filtered and dried overnight in a vacuum desiccator at 60 °C. In the second step, a dispersion of SEBS in ethanol (100 mL) was added to the iPP/G or iPP/GPP mixture, the dispersion was placed in an ultrasonic bath for 1 h and then the solvent was slowly removed at 90 °C on a heating plate. The material was dried at 60 °C in a desiccator. The nanocomposite formulation was tuned to 20 wt.% SEBS and 5 wt.% graphene, in order to achieve materials with a reasonably balanced toughness, stiffness, tensile elongation, and electrical conductivity.

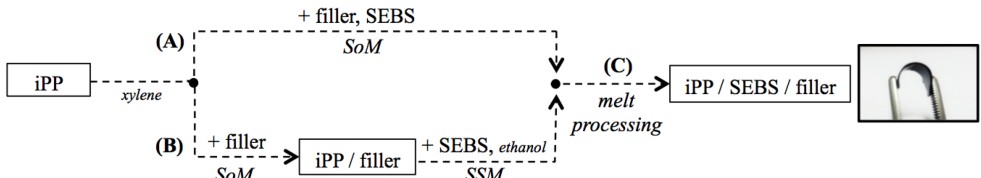

**Figure 1.**　　Formulation strategies for the preparation of isotactic polypropylene (iPP)/styrene–ethylene–butylene–styrene (SEBS) nanocomposites with graphene (G) or polypropylene chains to graphene (GPP) filler. (A) solution mixing (SoM) route, R1 and (B) solution/solid state mixing (SoM/SSM) route, R2.

All nanocomposites were melt-blended in a Haake Minilab extruder for 5 min at 210 °C and a rotor speed of 100 rpm. The material was then processed into films (thickness 0.5 mm) by hot-compression. A brass frame, in between two flat plates of the same material, was employed to control dimensions and guarantee uniform film thickness.

### 2.3. Characterization

TGA was carried out on the materials prior to and post extrusion, as well as on the thin films (Q-50, TA Instruments, New Castle, DE, USA), heating the sample (10 mg) from 50 °C to 800 °C at 10 °C/min under a nitrogen purge gas flow of 60 cm$^3$/min. Scanning electron microscopy (SEM) images were taken on a SU8000 Hitachi microscope (Tokyo, Japan) on cryo-fractured samples. The distribution of SEBS domains and the filler in the PP matrix was analyzed after submerging the samples in xylene for 12 h, washing the sample with the same solvent, and drying under vacuum overnight. The tensile properties were measured on an MTS QTest 1/L instrument (specimen dimensions: length 35 mm, width of grip 12 mm, width 2 mm, thickness 0.5 mm) at room temperature. Resistivity measurements were conducted with a four-point probe setup on films with a DC low-current source (LCS-02) and a digital micro-voltmeter (DMV-001) from Scientific Equipment & Services, Roorkee, India. The variation of resistance in a two-point probe configuration was monitored with a multimeter whilst manually bending the samples. The specimens (42 mm × 15 mm and thickness of 0.4 mm or 0.2 mm) were contacted to the circuit via Ag-ink painted electrodes.

### 3. Results and Discussion

As mentioned in the experimental part, two processing routes and two types of filler were tested in order to determine the optimal conditions for improved mechanical and electrical properties. The solution approach, in principle, guarantees more intimate interactions between all components. In addition, the use of graphene modified with short polypropylene brushes (GPP) was intended to selectively locate the filler in the iPP domains. The effect of the processing methods and the type of fillers on the macroscopic properties of the mixtures is described below.

### 3.1. Morphology

The morphology of cryo-fractured samples was analyzed by SEM after extraction of the SEBS component with xylene. The voids observed in the images indicate the sites where the SEBS was initially located. The control samples show a more homogeneous distribution of the SEBS domains throughout the inspected area when prepared via R1 (Figure 2A,B and Figure 3A,B and Figure S1).

This was expected since mixing in solution assures a better contact and interfacial interaction between components. At higher magnifications, the images obtained for R1 suggest the presence of more agglomerates in the sample prepared with pristine graphene (Figure 2C,D with respect to the sample with GPP (Figure 2E,F)). Premixing the filler with iPP via R2 reduces to some extent the aggregation of G and confirms an optimum exfoliation and homogeneous integration of the GPP filler within the matrix (Figure 3C–F).

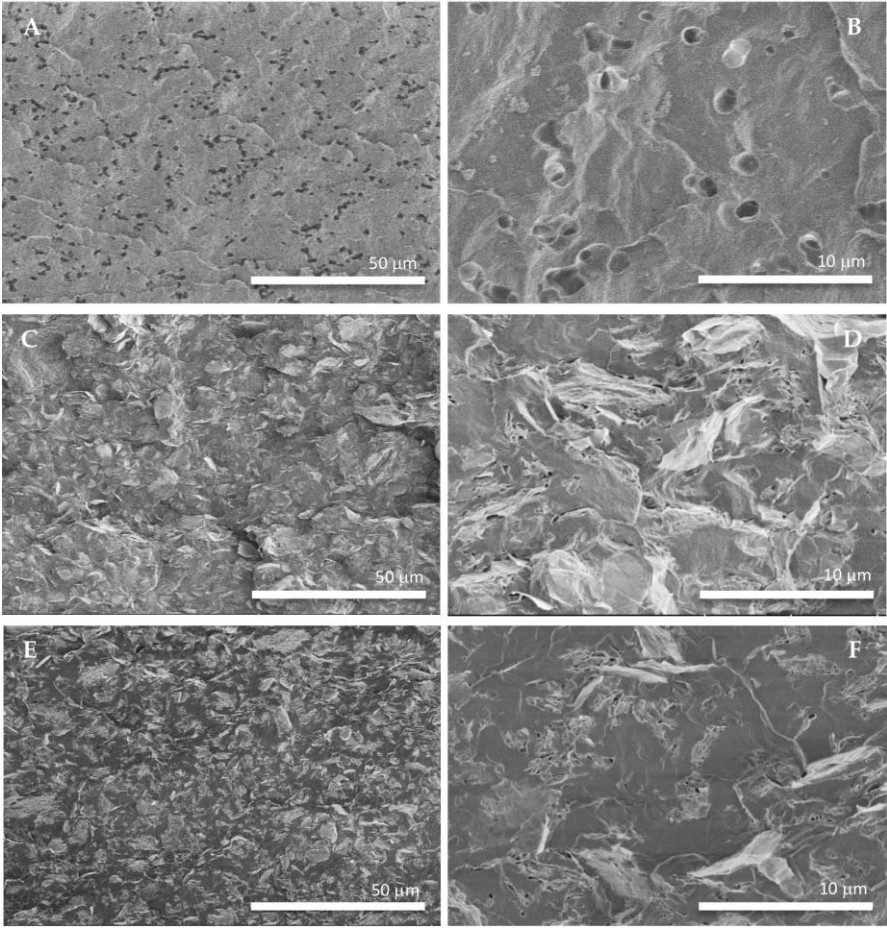

**Figure 2.** SEM images of cryo-fractured samples of iPP/SEBS (**A**,**B**); iPP/SEBS/G (**C**,**D**); iPP/SEBS/GPP (**E**,**F**); prepared via the SoM route. Scale bars in (**A**,**C**,**E**) correspond to 50 μm), while those in (**B**,**D**,**F**) correspond to 10 μm.

The analysis of the cavities that correspond to the volume previously occupied by the SEBS phase exhibit average domains sizes of 1.2–1.5 μm (Figure S1). The incorporation of G and GPP filler into the blend resulted in a significant decrease of the size of the cavities in all nanocomposites by approximately 80%, indicating a better dispersion of the elastomeric phase throughout the PP matrix, driven by the filler (Figure S1). Similar results have been observed previously for iPP/SEBS blends reinforced with montmorillonite [21]. In particular, the mean of cavity diameter is slightly higher when GPP is used as filler. A statistical analysis with Image J software shows cavity mean area of ~0.20 μm$^2$ and 0.25–0.31 μm$^2$ for G and GPP, respectively. In principle, the smaller the voids of SEBS domains, the better the impact resistance of the blends (see discussion below). It is interesting to note that beyond the changes in size, the morphology of the voids was also affected. While in almost all cases the shape of the voids looked circular, in the GPP sample they had an elongated shape and were found along the graphene surface (Figure S1). This can be attributed to the high affinity of graphene to

the styrene blocks of the SEBS domains through π–π interaction. However, the enlarged morphology can have a negative influence on the mechanical properties, as will be discussed later.

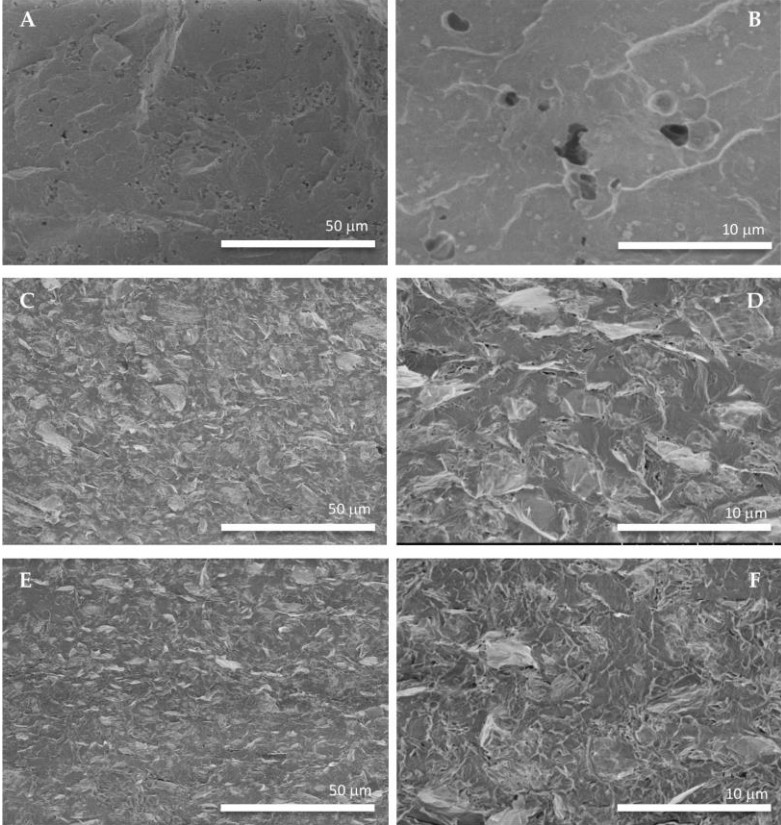

**Figure 3.** SEM images of cryo-fractured samples of iPP/SEBS (**A,B**); iPP/SEBS/G (**C,D**); iPP/SEBS/GPP (**E,F**) prepared via the SoM/SSM route. Scale bars in (**A,C,E**) correspond to 50 μm while those in (**B,D,F**) correspond to 10 μm.

### 3.2. Thermal Properties

TGA analysis was conducted to evaluate the homogeneity of the sample, to calculate the filler content and to monitor the stability of the materials throughout the compounding and processing steps (mixing, extrusion, and pressing). TGA curves of the materials prepared by both routes display only one degradation process (Figure 4). The observed degradation temperatures and char residue percentages are presented in Table 1. The influence of the formulation route was primarily observed for the iPP/SEBS blend, which shows improved thermal stability when all components are coprecipitated from solution (R1). The addition of G and GPP fillers improves the thermal properties, resulting in materials with very similar thermal stability. This confirms a good filler dispersion, providing optimum inhibition in the emission of pyrolytic gases a and delay in the matrix degradation. The results suggest little effect of the processing method and the type of filler used, as in all cases the ternary mixtures are more stable than the binary iPP/SEBS ones. The corrected residual weights from the TGA analysis verified the comparable filler loading in all nanocomposites, with values ranging between 4.8–5 wt.% G. No significant thermal degradation throughout the extrusion and melt-pressing steps was observed for all the materials studied.

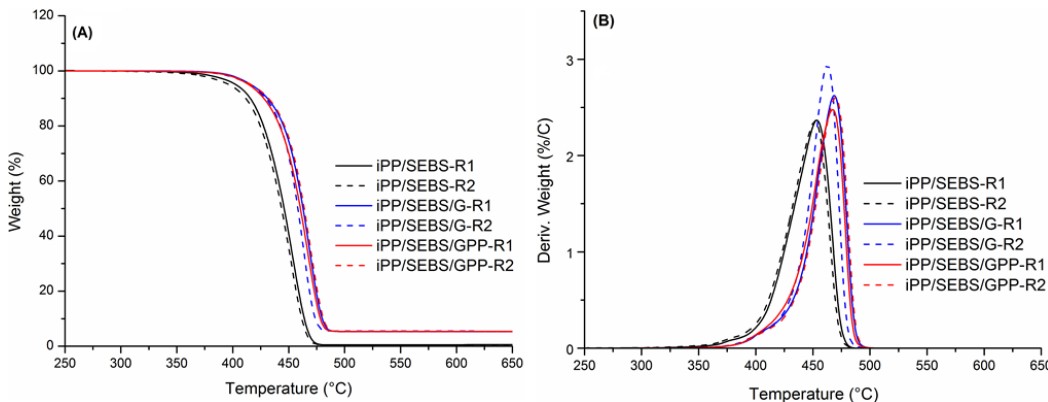

**Figure 4.** Thermogravimetric (TG) (**A**) and first derivative thermogravimetric (DTG) (**B**) curves of films prepared from iPP/SEBS composites with and without G and GPP filler obtained through the SoM and SoM/SSM formulation routes.

**Table 1.** Characteristic degradation temperatures and char residues obtained from the thermogravimetric analysis (TGA).

| Method | Sample | $T_i$ (°C) | $T_{max}$ (°C) | Residue (%) |
|--------|--------|--------|--------|--------|
|  | iPP/SEBS | 384.5 | 453.6 | 0.5 |
| R1 | iPP/SEBS/G | 401.9 | 468.7 | 4.8 * |
|  | iPP/SEBS/GPP | 400.1 | 467.4 | 5.0 * |
|  | iPP/SEBS | 376.1 | 450.0 | 0.6 |
| R2 | iPP/SEBS/G | 400.5 | 464.9 | 4.9 * |
|  | iPP/SEBS/GPP | 400.7 | 469.7 | 4.8 * |

$T_i$: Initial degradation temperature obtained at 2% weight loss, $T_{max}$: Temperature of maximum weight loss rate, * corrected value with blank sample residue.

## 3.3. Mechanical Properties

The iPP/SEBS blend and nanocomposites prepared via both routes were subjected to tensile testing measurements. The Young's modulus, elongation at break, tensile strength, and toughness values are listed in Table 2 (see stress–strain curves in Figure S2). Firstly, the iPP/SEBS blend prepared by R1 outperformed in stiffness (Young's modulus), deformation capacity (elongation at break) and toughness when compared to the same sample prepared by R2. This can be explained by the greater homogeneity of the composite and the smaller SEBS domains when formulated via R1, in agreement with the SEM data. It is worth highlighting, that the latter sample exhibits extremely large deformations and shows the phenomenon of strain hardening with an ultimate stress of 25.8 MPa (Figure S2A). This effect can be related to the more uniform and superior stretching behavior, toughness, and ductility achieved in the more homogeneous sample. As a result, the iPP molecules can align in the direction of the load at larger strains. The Young's modulus gradually increases with the addition of G and GPP fillers for both routes, as typically observed for graphene nanocomposites. The addition of the fillers counteracts the improvement of the elongation capacity achieved by the SEBS additive, this being more marked for samples prepared by R1. On the contrary, the decrease in the tensile strength derived from the elastomeric component is better balanced with the addition of G and GPP when the composite is prepared via R2. Especially interesting is the variation of toughness, since it was maximum in binary mixtures prepared by R1 and becomes almost null when the fillers are added. However, in the case of R2, toughness also decreases with the addition of graphene albeit maintaining reasonably good values compared with iPP. Within R2 approach, the best stiffness/toughness relationship was found for the sample with unmodified graphene. Although the better dispersion of GPP in the iPP matrix is expected to improve properties, this effect can be counteracted by the formation of non-cylindrical and larger SEBS domains (as seen by SEM). In addition, the location of the rigid filler at the SEBS/iPP interfaces

could alter the elastomeric properties of SEBS, which translate into lower values of toughness and elongation at break. This effect is primarily observed for the R1, where the homogenous solution of polymers and the filler dispersion may favor both π–π as well as olefinic interactions simultaneously. It is also worth noting that the addition of the GPP filler generates the material with the greatest resistance to deformation if prepared via R2, where the interfacial interaction between the polyolefins from the matrix and filler are likely to reinforce their adhesion (8% improvement compared to the optimum value for the iPP/SEBS composite). Thus, the addition of SEBS to iPP/G composites via the SoM/SSM approach imparts well-adjusted properties of stiffness, toughness, and elasticity to the graphene-reinforced iPP matrix.

**Table 2.** Mechanical properties of iPP and composites (iPP/SEBS/G and iPP/SEBS/GPP) determined from tensile tests and SEM area analysis.

| Method | Sample | E (MPa) | $\varepsilon_b$ (%) | σ (MPa) | T (MJ·m$^{-3}$) | SEBS Mean Area (μm$^2$) |
|---|---|---|---|---|---|---|
|  | iPP * | 660 ± 30 | 11.5 ± 1.0 | 24.0 ± 7.0 | 1.3 ± 0.2 |  |
| R1 | iPP/SEBS | 652 ± 31 | 450.1 ± 28.0 | 21.7 ± 1.7 | 97.3 ± 6,4 | 1.20 ± 0.20 |
|  | iPP/SEBS/G | 693 ± 7 | 5.3 ± 1.3 | 23.1 ± 1.5 | 0.9 ± 0.1 | 0.20 ± 0.03 |
|  | iPP/SEBS/GPP | 701 ± 22 | 4.2 ± 0.5 | 20.9 ± 1.0 | 0.6 ± 0.1 | 0.25 ± 0.04 |
| R2 | iPP/SEBS | 622 ± 27 | 55.0 ± 8.0 | 21.0 ± 0.4 | 11.1 ± 1.1 | 1.50 ± 0.30 |
|  | iPP/SEBS/G | 690 ± 9 | 17.2 ± 2.5 | 26.6 ± 0.6 | 4.0 ± 0.5 | 0.20 ± 0.03 |
|  | iPP/SEBS/GPP | 704 ± 22 | 10.0 ± 2.3 | 23.6 ± 0.8 | 1.9 ± 0.4 | 0.31 ± 0.05 |

E: Young's modulus, $\varepsilon_b$: Elongation at break, σ tensile strength, T: Toughness. * Data from previous studies [10].

## 3.4. Resistivity Measurements

The conductivity of the nanocomposite films was measured by the 4-point probe method and the values for each sample are listed in Table 3. The highest conductivity values were obtained for the sample prepared with the G filler, reaching $5.2 \times 10^{-3}$ S/cm and $6.8 \times 10^{-3}$ S/cm for R1 and R2, respectively. The samples prepared with the GPP filler are slightly less conductive with respect to the graphene counterpart. This is expected as a consequence of the grafting of an insulating moiety and the introduction of new defects to the sp$^2$ network of graphene. A substantial reduction of conductivities has been previously measured for GPP pellets when compared to the pristine G [10]. Additionally, the constant conductivity value of iPP/SEBS/GPP for both routes can be attributed to the good polymer/filler interface established, independent of the formulation route, as demonstrated by SEM. On the other hand, it is worth noting that the conductivity values obtained for all samples in this study were higher than values previously reported for nanocomposites with comparable filler contents: PP/SEBS/G-nanoplatelet (GN) (~$10^{-5}$ S/cm, 5 wt.% GN) [12] prepared through melt compounding, iPP/G ($8.7 \times 10^{-3}$ S/cm, 6.1 wt.% G), or iPP/GPP ($1.2 \times 10^{-3}$ S/cm, 4.6 wt.% G) [4] prepared via SoM and melt compounding.

**Table 3.** Electrical conductivities (in S/cm) of iPP/SEBS/G and iPP/SEBS/GPP calculated from the 4-point probe tests.

| Sample | SoM | SoM/SSM |
|---|---|---|
| iPP/SEBS/G | $5.2 \times 10^{-3}$ | $6.8 \times 10^{-3}$ |
| iPP/SEBS/GPP | $1.9 \times 10^{-3}$ | $1.9 \times 10^{-3}$ |

Furthermore, the electrical resistance of these samples was observed to vary upon bending the composite films (Figure 5). For example, the resistance for a iPP/SEBS/G film (thickness of 0.4 mm) increased with increasing degree of flexion, and returned to the original value upon removal of the external stress, even after repeated cycles. This observation can be explained by pulling apart

the graphene sheets while applying the external force, thus weakening their contacts and thereby increasing the resistance.

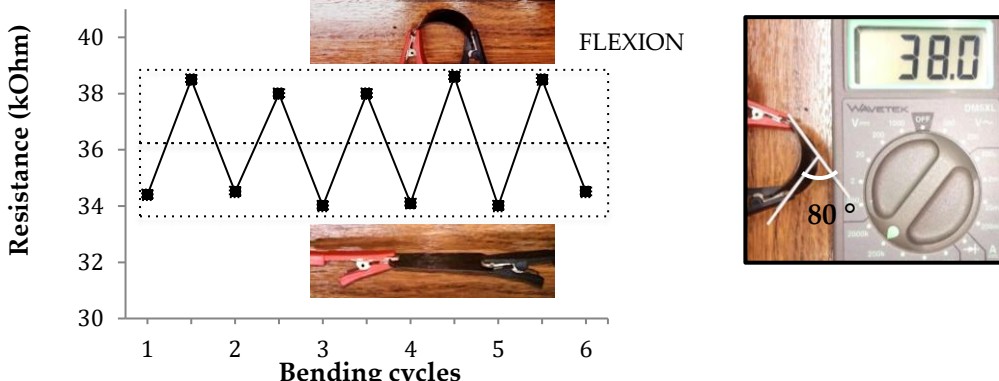

**Figure 5.** Resistivity response for iPP/SEBS/G film during dynamic bending cycles (**left**) and photograph of a film flexed about 80° (**right**).

## 4. Conclusions

Ternary mixtures of iPP/SEBS/graphene with piezoresistive properties were prepared, with the effect of the processing method and the nature of the filler employed on the final properties of the nanocomposites analyzed. It is concluded that the mechanical properties of the iPP/SEBS blend are critically influenced by the formulation method. The solution-blending approach afforded a material with the best viscoelastic properties because the coprecipitation of both materials allowed a homogenous distribution of the elastomeric domains throughout the iPP matrix. On the other hand, when functionalized or unmodified graphene was incorporated, the pre-blending of the fillers with the thermoplastic iPP emerges as the best approach as it generates materials with a good balance of mechanical properties and electrical conductivity. The nature of the filler exerts little effect on the final macroscopic properties of the nanocomposites. The graphene fillers (unmodified or modified) act as compatibilizers between the two immiscible phases of iPP and SEBS providing improved electrical and mechanical properties when the two-step procedure is employed. The nanocomposites obtained presented a good balance between stiffness, flexibility, toughness, and electrical conductivity, not previously observed for similar materials. In this work, the reinforced blends prepared by the SoM/SSM route provides composites with improved electrical conductivity while retaining the mechanical properties of iPP, making them attractive materials for low cost and high throughput flexisensors and actuators, making them potential candidates for components in the automotive sector.

**Supplementary Materials:** The following are available online at http://www.mdpi.com/2504-477X/3/2/37/s1, Figure S1: Additional SEM images of cryo-fracture samples; Figure S2: Stress-strain curves for all samples studied.

**Author Contributions:** Conceptualization, H.J.S. and M.A.G.-F.; investigation, H.S.; validation, H.S. and H.J.S.; writing—original draft preparation, H.S.; writing—review and editing, H.J.S. and M.A.G.-F.; visualization, H.S.; funding acquisition, H.J.S. and M.A.G.-F.

**Funding:** This research was funded by the Spanish government (MINECO), grant number MAT2017-88382-P.

**Conflicts of Interest:** The authors declare no conflict of interest.

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
