# Peer review of "Preparation of Piezo-Resistive Materials by Combination of PP, SEBS and Graphene"

_jcs, doi:10.3390/jcs3020037_

Round 1

Reviewer 1 Report

In this paper, the preparation of polypropylene/styrene–ethylene–butylene–styrene triblock copolymer/graphene system directed to materials combining better mechanical properties and electrical conductivity was described. Graphene and polypropylene‐modified graphene were used as fillers. Previous to melt processing, two routes were used to prepare the blends: a solution mixing (SoM) and a solution/solid state mixing (SoM/SSM). The nanocomposites prepared via the SoM and SoM/SSM have been characterized by SEM, TGA and the electrical and mechanical properties have been evaluated.

The article presents innovation in the synthesis of nanocomposites and makes a well-documented study about its morphological and thermal features. The relationship between these characteristics and the mechanical properties is well explained. The study has shown interest, especially in the area of application of the composite with these characteristics.

°         A more careful writing is necessary because there are some spelling mistakes:

·         Pag. 3 line 96: “achive”;

·         Pag. 3 line 108: “materiales”;

·         Pag. 6 line 200: “proprierties”;

·         Pag. 6 line 209: “unifm”;

·         Pag. 7 line 210: “streching“, “behaivoir” and “homogeneuous”;

·         Pag. 7 line 222: “Althougth“;

·         Pag. 8 line 250: “smaples“.

°         In the paragraph beginning at line 254, the authors need to mention figure 5, since the results refer to that figure.

°         In table 3, it is necessary to indicate the units of electrical conductivities.

Author Response

We thank the reviewer for this positive feedback on our article. We have addressed all the points raised by the reviewer. In addition, the language has been corrected by a native English speaking colleague. Consequently, the text has been substantially improved.

Reviewer 2 Report

Please find comments regarding the manuscript below:

The introduction doesn't provide sufficient background, the references section is rather poor. Please update this section.

The SEM pictures should have scale bar placed on the picture, in the present form it is misleading. Please provide better quality pictures and description.

Author Response

We thank the reviewer for these helpful comments.

The introduction section has been rewritten and twelve additional references have been incorporated.

Better quality SEM images, all containing scale bar have been added to the manuscript.

Reviewer 3 Report

This manuscript attempted to address electrical properties of hybrid PP, SEBS and graphene composite. The manuscript lack seriously the academic integrity, flow and novelty. The purpose of the work is well established. Given the very little data and study, their manuscript is incomplete. Specific comments are given below. For these reasons, I reject this submission at this time.

1.       Novelty of this particular work should more efficiently conveyed. What adds new knowledge from this study compared to existing literature?

2.       The general style of the paper is quite informal and somewhat clumsy. This is not recommended in a journal publication, not only from a stylistic point of view but also (and more importantly) as it negatively impacts the clarity of the paper. Again my main concern was, the flow between each section is somewhat disjointed.

3.       Several occasion, the results are not explained with their implications; by this, in opinion, the manuscript is incomplete. Results and discussion part is indeed very short as result of very little data.

4.       Characterisation section is indeed insufficient with missing experimental conditions.

5.       Formatting and language

a.       There are so many spelling errors and tenses change abruptly in several places. Punctuations are poorly followed that mislead several important statements and arguments.

b.      Serious attention should be given to grammatical, singular/plural and prepositions mistakes throughout the manuscripts.

c.       Not all the acronyms are expanded in the abstract or acronyms are improperly followed.

d.      Use consistent number of digits in values.

e.      Expression of several facts is not quite scientific.

6.       Images are very poorly presented.

a.       SEM images are not in high quality which restricts the clear visualisation of the microstructure.

Author Response

We would like to express our total disagreement with the reviewer's statements. The sentence: “The manuscript lack seriously the academic integrity” is not professional without justification. However, we have addressed the issues raised by the reviewer in order to improve the manuscript.

Our point-by-point replies follow:

1- Regarding novelty, we have found only three papers on similar systems as the reported here, which have been incorporated to the revised version. Among these, two papers entirely focused on mechanical properties and one reported electrical conductivity values. In our study, the use of chemically modified graphene allows us to direct the nanofillers to specific blend domains. In this way, the elastomeric properties are maintained, while electrical conductivity values higher than the reported up to now are achieved. In addition, the electrical conductivity remains unaltered under bending cycles, allowing monitoring of mechanical changes. In our modest opinion, using specific chemistry on graphene and the reported blending protocol to prepare materials that meet acceptable mechanical properties with good and stable electrical conductivity can be considered "new knowledge".

2- The language and style has been deeply revised by a native English speaking colleague and the manuscript is now easier to follow.

3- As the reviewer is aware the key aspect in polymer nanocomposites is related to the distribution of the filler in the matrix. SEM is the ideal technique to assess this point and SEM results are included and discussed in this manuscript. Regarding properties, thermal stability, mechanical, electrical and "electromechanical" properties have been evaluated. So, we do not consider the manuscript contains "very little data". Furthermore, an interpretation for all the variations observed in each property is given (maybe the referee prefers long and tedious explanations but we prefer to get to the point).

4- We think we have included all the experimental details. It is difficult to modify this section without having details from the reviewer about what information is missing. We have tried to guess and we have included some additional information that we believe may be relevant.

5- We thank the reviewer for these comments. We have modified the manuscript accordingly. As mentioned before, the language has been revised by a native English speaking colleague.

6- Better quality SEM images have been included in the revised version.

Round 2

Reviewer 3 Report

The revised version improved to an extent. However, the following should be given attention.

Figure 1 is not referred to in the text.

SEM scale bar should be given.

TGA- thermograms should be clearly presented. Derivative loss curves should be given separately or the other one.

English is not still satisfactory.

Author Response

The manuscript has been revised according to the reviewer's recommendations.

- A reference to Figure 1 has been included in page 3, second paragraph

- Scale bars for each SEM images have been  included

- Figure 4 has been changed and TGA and DTGA curves are now shown separately

-No further action have been taken with the language because the present version of the manuscript was corrected by a English native colleague